# Suitability of Auger Pressing Briquettes for Blast Furnace Use Based on Laboratory Tests

**Olli Vitikka** [1,*], **Mikko Iljana** [1], **Anne Heikkilä** [1], **Illia Tkalenko** [2], **Nikita Koriuchev** [2], **Daniel Shehovsov** [2], **Andrey Malkki** [3] **and Timo Fabritius** [1]

1   Process Metallurgy Research Unit, University of Oulu, P.O. Box 4300, FI-90014 Oulu, Finland; mikko.iljana@oulu.fi (M.I.); anne.heikkila@oulu.fi (A.H.); timo.fabritius@oulu.fi (T.F.)
2   AMCOM GROUP LLC, 34 Saint James Drive, Palm Beach Gardens, FL 33418, USA; ilt@amcom-intl.com (I.T.); n.koryuchev@amcom-intl.com (N.K.); daniel@amcom-intl.com (D.S.)
3   Kivisampo Oy, Tehontie 45, FI-45200 Kouvola, Finland; andrey.malkki@kivisampo.fi
*   Correspondence: olli.vitikka@oulu.fi

**Abstract:** Briquetting is a process in which fine materials unsuitable for use as such are agglomerated to achieve a larger particle size. Auger pressing is a novel briquetting method to efficiently improve the recycling of by-products from iron and steelmaking. The high-temperature properties of auger pressing briquettes mainly consisting of blast furnace sludge and mill scale were evaluated. The aim was to determine the suitability of the briquettes for blast furnace (BF) ironmaking by studying the reduction, swelling, and cracking behavior using a laboratory-scale furnace. The blast furnace simulator (BFS) capable of performing non-isothermal reduction experiments with changing gas compositions was used to simulate the different stages of reduction up to 1100 °C in an atmosphere with $N_2$, $CO$, and $CO_2$ gases. A commercial olivine pellet and a conventional industrial BF briquette were used as reference samples. The sample weight losses were monitored by thermogravimetry, swelling as a change in the volume, and cracking by visual inspection. The samples were analyzed using microscopes and an elemental analyzer. Based on the BFS experiments, the briquettes proved to be a promising raw material for BF use. They were of a self-reducing quality due to their carbon content and showed reduction to metallic iron faster compared to the reference samples. The swelling was slight, and despite the minor cracking the structure of the briquettes did not degrade.

**Keywords:** auger pressing; briquettes; by-products; reduction; blast furnace; ironmaking

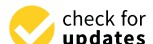



## 1. Introduction

Blast furnaces (BF) play a major role in steelmaking. In 2020, the blast furnace–basic oxygen furnace (BF-BOF) process route covered 73.2% of world crude steel production [1]. In addition to iron and steel, various side streams are generated in the process chain. For every ton of hot metal produced, about 400 kg of by-products including different kinds of slags, scales, dusts, and sludges are generated in the BF-BOF route. This is twice as much as the electric arc furnace (EAF) route, which is an alternative steelmaking process [2,3]. A significant part of the residues is recycled in-plant via sintering, cold bonded agglomeration, or further as direct reduced iron (DRI). For example, iron and steelmaking slags can be used in binders as a partial substitute for cement and, due to their properties, they are sold to the cement industry and for road construction. Moreover, many by-products including EAF dust and highly carbonaceous BF dust and BF sludge have a high iron content which makes recycling feasible [4–6]. Furthermore, recycling significantly reduces the landfill of products, which saves natural resources and prevents risks to human health and the environment [7,8].

One of the key challenges in recycling iron and steelmaking by-products is their small particle size. The permeability of the ore layer is determined by the amount of fines (below 5 mm). A decrease in permeability may cause operational problems and affect the

productivity of the BF process [9,10]. The material to be recycled, such as BF sludge [11], can also be moist, which further complicates the detection of fines [9]. Therefore, a suitable agglomeration technique is needed. Another issue relevant to recycling is the chemical composition of the by-products. In addition to valuable elements enabling self-reduction, i.e., iron oxides and carbon in the form of coke, the side streams often contain undesired or harmful elements. For example, BF sludge may contain considerable amounts of zinc and alkali compounds. The evaporation and condensation behavior of zinc hinders the BF operation, while alkalis impair the hot strength of coke and weaken the refractory lining, which shortens the campaign length [6,9,12,13]. Thus, the harmful contents of the products must be considered and, if necessary, disposed of before recycling.

Cold bonded briquetting is a well-known agglomeration method in which a binder, often bentonite, is added to a mixture of fine-grained materials to be recycled. There are three basic briquetting technologies: vibro-pressing, roller pressing, and stiff extrusion [14]. Vibro-pressing or punch-and-die pressing is the oldest high-pressure agglomeration technology and is used in various industries. In addition to steelmaking, it is used in ceramic, powder metal, and the general chemical industries [15]. The Hess Group has been one of the implementers of such solutions [16]. In this work, a briquette produced by this technology is used as a reference sample. Another high-pressure agglomeration technology, roller pressing, is also used in briquetting. For example, Lemos et al. [17] studied the behavior of roller pressing briquettes made of BF dust and sludge. In the field of roller pressing, Köppern is one of the market leaders in the manufacture and supply of machinery and plants for the processes of briquetting, compaction, and comminution [18]. The third technique, i.e., extrusion technology, was first applied in an industrial scale by J.C. Steele & Sons in 1993 at the metallurgical plant of Bethlehem Steel Corp. in Pennsylvania, USA [14].

The extreme conditions of the BF process place a wide range of requirements for the raw materials. These include sufficient reducibility, high cold strength, low reduction-disintegration index (RDI), slight variations in chemical composition, and suitable particle size [17]. These relate to the three main metallurgical properties under consideration in this work: reducibility, swelling, and cracking. Softening and melting in the cohesive zone where the temperatures vary between 1000 and 1350 °C have been studied especially in the case of iron ore pellets [19] but are not considered in this study because the temperatures to be simulated would be too high. Reducibility refers to the ability of a material to increase the extent of reduction reactions while indicating both the ratio of direct reduction and the heat consumption at the lower zone of the BF [9,20]. Swelling, especially in the case of pellets, has been extensively studied and has been found to depend mainly on the basicity and gangue content of the material. Swelling may be normal, which is advantageous for the reduction process due to increased porosity of the material, or abnormal, which may cause operating problems due to the deterioration of the mechanical properties of the material. Cracking is often associated with reduction and swelling behavior and occurs at higher degrees of reduction [9,21].

Unlike various iron burden materials such as sinter, lump ore, and pellets that can be characterized by the means of size, cold strength, and reducibility [6], briquettes are not mentioned in the ISO standards on quality requirements. The reducibility tests described in ISO 4695 [22] and ISO 7215 [23] use constant temperatures (900 °C ± 10 °C or 950 °C ± 10 °C) throughout the test. Since it has been found that the behavior of iron ore pellets in the laboratory and in the BF differs due to the complex reducing and melting conditions of the BF process [24], non-ISO standardized tests are designed to simulate the actual BF conditions [21].

In this work, the high-temperature properties of auger pressing briquettes made of BF sludge and mill scale, similar to the extrusion briquettes mentioned above, were evaluated using a laboratory-scale furnace. It was of particular interest to determine whether the briquette is of self-reducing quality and how rapidly the reduction occurs, considering external changes during the reduction under simulated BF conditions. The aim was to

obtain information on the suitability of briquettes for BF use in order to assess the potential of this briquetting process for the recycling needs of steelmaking.

## 2. Materials and Methods

### 2.1. Auger Pressing Briquettes

The briquettes studied in this work are cold-bonded agglomerates produced utilizing vacuum auger pressing technology developed by AMCOM GROUP LLC. The technology allows the use of finely dispersed natural and technogenic materials obtained from steel mills and other industries. The process consists of preparation processes including box feeders, the double shaft mixer with a binder feeder, and the mixer-press with water dosing. The vacuum auger pressing equipment is the main multi-component unit of the production line. It is equipped with water dosing and produces briquettes which are aged for three days to achieve optimum strength. The unit shown in Figure 1 consists of three functions:

1.  Mixing chamber including double shaft mixer and pre-compaction function;
2.  Vacuum chamber;
3.  Auger press chamber.

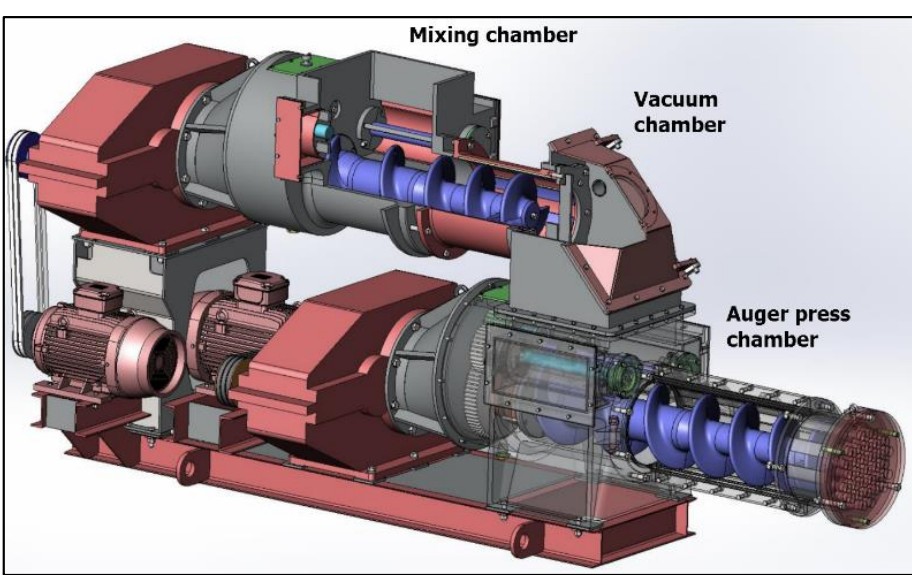

**Figure 1.** Technological scheme of a vacuum auger pressing unit.

The auger press chamber consists of three zones with decreasing diameters towards the end of the chamber: the mixing zone, the pre-compression zone, and the compression zone. The rotating auger forces the briquette paste through the holes in the die, as a result of which the briquette becomes cylindrical. The briquette breaks to its final length due to gravity. Productivity has been reported to be from 3 to 90 t/h.

Laboratory tests of customer materials and binder selection are carried out in AMCOM GROUP LLC laboratories located in Europe. The production takes place in locations where raw materials are available. In Finland, production is carried out by Kivisampo Oy in Kouvola. Various iron and steelmaking products such as ferrosilicon, ferrochrome, mill scale, BF dust, and EAF gas cleaning dust have already been utilized. The binders used in briquettes are both inorganic and mixtures of inorganic and organic materials and do not contain harmful impurities. Since 2019, three industrial briquetting lines have been established in CIS countries. BF sludge, EAF dust, and mill scale were used as raw materials with 1%–2% of mineral-organic binder. In 2022, auger pressing briquettes made from mill scale were used as a charge material in a successful test using an industrial BF with a volume of 2002 $m^3$ in Mariupol, Ukraine. No harmful effects were detected.

Briquettes used in this work were produced from iron and steelmaking by-products. Thirty kilograms of briquettes were manufactured in the AMCOM GROUP LLC laboratory

based on the briquette recipe shown in Table 1. As seen from the calculated composition in Table 1, the main constituents of the briquettes were BF sludge (40%), mill scale (55%), slaked lime (3%), and binder (2%). The moisture content before adding water was reported to be 1.27%.

**Table 1.** Main constituents, their contents, and elemental analysis in auger pressing briquette recipe.

| Constituents | Elements (%) | | | | | | | | | Weight (g) | Content (%) |
| | $Fe_{tot}$ | P | $SiO_2$ | CaO | MgO | $Al_2O_3$ | Zn | $H_2O$ | C | | |
|---|---|---|---|---|---|---|---|---|---|---|---|
| BF sludge (0–3 mm) | 44.4 | 0.17 | 5.12 | 3.12 | 3.9 | 2.2 | 0.43 | | 23.0 | 12,000 | 40 |
| Mill scale (0–3 mm) | 70.0 | | | | | | | | | 16,500 | 55 |
| Slaked lime (0–3 mm) | | | | 80.0 | | | | 20.0 | | 900 | 3 |
| Binder AMCOM VA3500 | | | | | | | | | 33.3 | 600 | 2 |
| Calculated Total Content | 56.3 | 0.07 | 2.05 | 3.65 | 1.56 | 0.88 | 0.17 | 0.60 | 9.87 | 30,000 | 100 |
| Added water | | | | | | | | 100.0 | | 4500 | |

### 2.2. Iron Ore Pellets

Iron ore pellets, spherical particles with a diameter of around 10 mm, are a general iron burden material. They are specially designed for extreme conditions in terms of cold strength, swelling, softening, and melting properties. Pellets are usually very rich in iron and increase the total iron content of the charge materials. Basicity and gangue content vary depending on the pellet type. The three main pellet types are acid pellets ($CaO/SiO_2$ ratio < 0.5), basic or fluxed pellets ($CaO/SiO_2$ ratio between 0.9 and 1.3) with limestone ($CaCO_3$) or dolomite ($Ca,Mg(CO_3)_2$) as additives, and olivine pellets with olivine ($Mg_2SiO_4$) as the fluxing additive. Olivine pellets contain magnesium oxide (MgO) instead of calcium oxide (CaO) [9,25].

Commercial olivine pellets were used as a reference sample in this study. The average weight of a 10–12.7 mm sized pellet was 3.4 g. The chemical analysis of the reference pellet is shown in Table 2. Calculated B2 basicity ($CaO/SiO_2$ ratio) of the pellet was 0.23.

**Table 2.** Chemical analysis of the reference pellet (wt.-%).

| $Fe_{tot}$ | FeO | $SiO_2$ | CaO | MgO | $Al_2O_3$ | $TiO_2$ | $V_2O_5$ |
|---|---|---|---|---|---|---|---|
| 66.7 | 0.6 | 1.85 | 0.43 | 1.3 | 0.32 | 0.35 | 0.26 |

### 2.3. Industrial Blast Furnace Briquettes

In order to compare the high-temperature properties of auger pressing briquettes with other by-product briquettes, the industrial BF briquette produced at the SSAB steel plant in Raahe, Finland was used in the experiments. The briquettes are used as one of the BF input materials. The briquette differs from the auger pressing briquette in terms of agglomeration method, shape, size, and chemical composition. The briquette is made by the punch-and-die (vibro-pressing) agglomeration method, weighs about 400 g, and its end is hexagonal in shape. The briquette consists mainly of mill scale, pellet screenings, steel scrap, BF dust, coke dust, and cement. Depending on the season, the cement is replaced by 30%–50% ground-granulated blast furnace slag (GGBFS) [26]. The chemical composition shown in Table 3 was received with the briquettes.

**Table 3.** Chemical analysis of the reference briquette (wt.-%).

| C | Na$_2$O | MgO | Al$_2$O$_3$ | SiO$_2$ | S | K$_2$O | CaO | Fe | Basicity |
|---|---|---|---|---|---|---|---|---|---|
| 8.28 | 0.32 | 2.1 | 2.4 | 8.1 | 0.41 | 0.21 | 11.1 | 48.4 | 1.37 |

*2.4. Strength Tests*

Metallurgical tests have been developed to characterize iron burden materials from the perspective of different metallurgical properties. These include cold strength, RDI, reducibility, softening, and melting properties. The tests, which are usually ISO-standardized, are generally used to characterize sinter and pellets [5] and were not intended for briquettes. Reducibility tests based on constant temperature according to ISO 4695 and ISO 7215 were not used in this work as the purpose was to simulate actual BF conditions.

Although briquettes differ from other iron burden materials, mechanical strength tests have been applied to briquettes in a study by Mousa et al. [18] in which was found that the behavior of briquettes differs from that of pellets due to the larger size of the briquettes. However, the tests provide useful information on the cold strength of the briquettes. Three different cold strength tests demonstrated in Figure 2 were performed on the briquettes studied in this work: mechanical crushing strength test, drop strength test, and abrasion strength test. These were carried out in the AMCOM GROUP LLC laboratory.

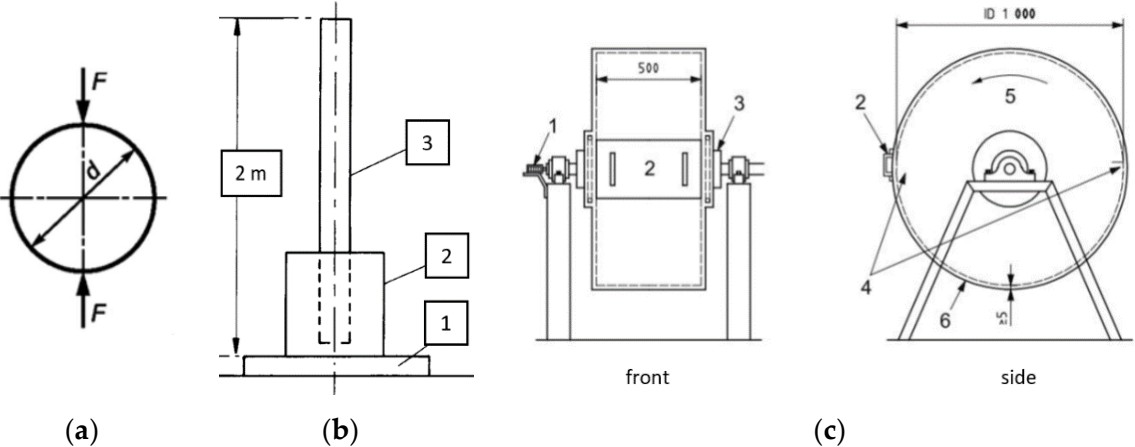

**Figure 2.** Sketches of strength test methods used: (**a**) scheme of loading in crushing strength tests; (**b**) drop strength test equipment with (1) heavy plate, (2) tube or collar, (3) vertical pipe [15], and (**c**) tumbler test drum with (1) revolution counter, (2) door with handle, (3) stub axle, (4) lifters, (5) direction of rotation, (6) plate [27].

The mechanical crushing strength of the briquettes was tested after 72 h of aging by crushing along the loading axis. The mechanical drop strength was tested by 3-fold dropping using a drop height of 2 m. The abrasion strength test was performed as a tumbler test in a rotating drum for a total of 200 revolutions.

*2.5. Sample Preparation*

A total of five auger pressing briquettes were prepared for the study by keeping them in a temperature cabinet at 105 °C overnight. This was done to avoid moisture absorption from air. The moisture content was calculated by weighing the samples before and after the treatment. The moisture content was approximately 1%. Four of the five samples were used in the reduction experiments, and one was retained as the original sample.

The iron ore pellets were screened using 10 and 12.5 mm sieves, and 35 round-shaped samples were selected from the screened pellets. The reference briquettes were large but still suitable for the test equipment without cutting into smaller pieces. Both reference pellets and reference briquettes were treated in the temperature cabinet in the same manner

as the auger pressing briquettes and weighed before the experiments. The auger pressing briquette, reference pellet, and reference briquette placed in the sample basket are shown in Figure 3.

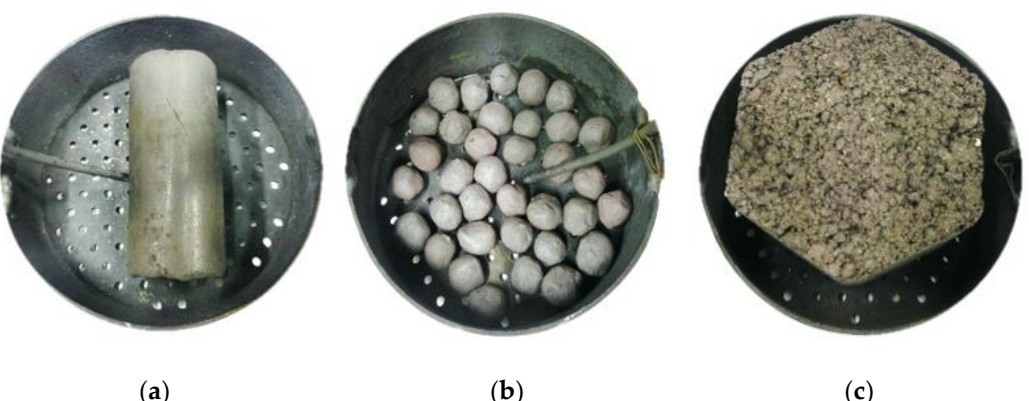

(**a**)　　　　　　　　　　　　　　(**b**)　　　　　　　　　　　　　　(**c**)

**Figure 3.** Sample basket pictured with the samples: (**a**) auger pressing briquette; (**b**) iron ore pellets; and (**c**) reference briquette.

### 2.6. Blast Furnace Simulation

The blast furnace simulator (BFS) presented in Figure 4 is a tube furnace first introduced by Iljana et al. [21] in a study on the reduction and swelling of iron ore pellets. The reduction tube is made of heat-resistant steel and has a diameter of 95 mm. The BF process can be simulated up to 1100 °C and $N_2$, CO, $CO_2$, $H_2$, $H_2O$, $S_2$, and K gases can be dispensed via pre-determined computer-controlled reduction programs which allow the creation of complex time and temperature-dependent atmosphere profiles. Changes during the reduction tests can be monitored through a video camera system with a mirror and light source.

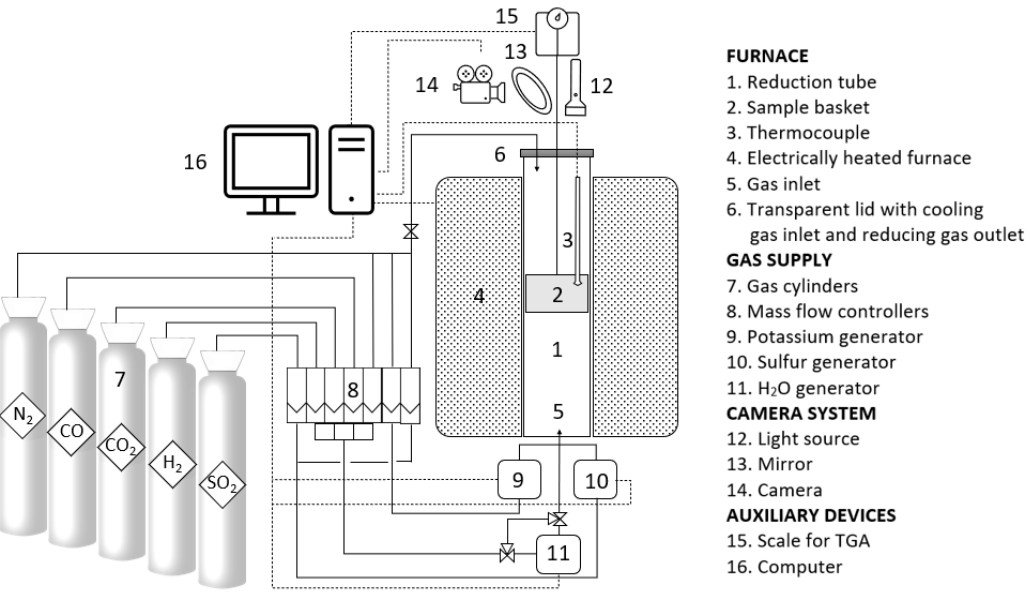

**Figure 4.** Operational scheme of blast furnace simulator (BFS).

Dynamic reduction experiments were performed in an atmosphere with $N_2$, CO, and $CO_2$ gases. The separate sulfur generator heated to 800 °C was utilized to pre-heat the gas before starting the reduction experiments. A 280 min reduction program, Experiment A, with a 40 min isothermal period at 1100 °C was created as seen in Figure 5. Moreover,

three shorter programs, referred to as Experiments B–D, were modified to simulate the different stages of reduction in the BF process. The set gas compositions throughout the Experiments A–D can be seen in Figure 5, presenting Experiment A as a function of time. The reducing conditions based on gas compositions during the experiments are presented as an Fe–O–CO–CO$_2$ phase diagram, often referred to as Bauer–Glaessner diagram, in Figure 6. Reduction experiments performed on iron ore pellets and reference briquettes are referred to as Experiment E and Experiment F. Experiment E used the program for Experiment B, while Experiment F used the program for Experiment A. That is, a 40 min isotherm was used for the reference briquette but not for the reference pellet. This was done to compare the reduction behavior of briquettes.

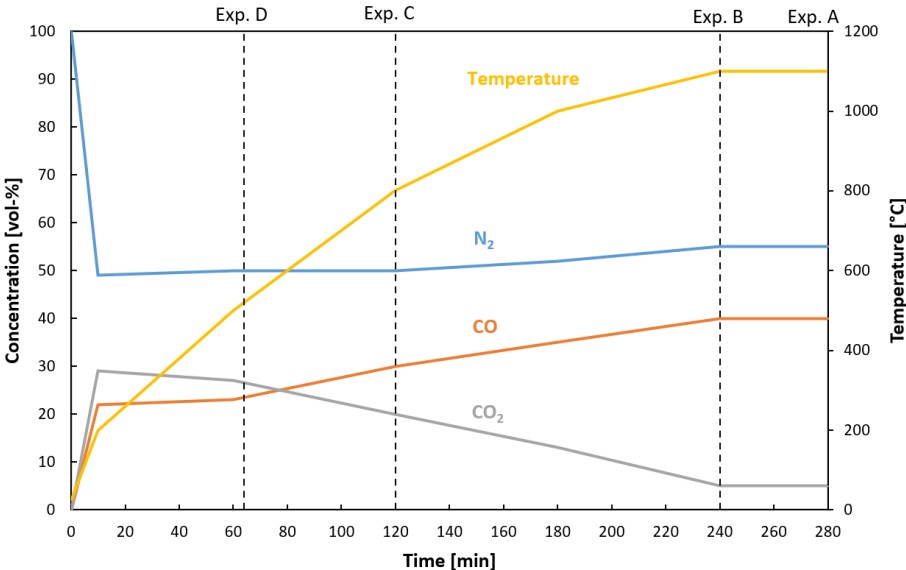

**Figure 5.** The programmed concentrations of N$_2$, CO, and CO$_2$ gases during the uninterrupted reduction experiment (Experiment A) with the 40 min isotherm. The heating endpoints for the shorter runs (Experiments B–D) are shown in dashed lines.

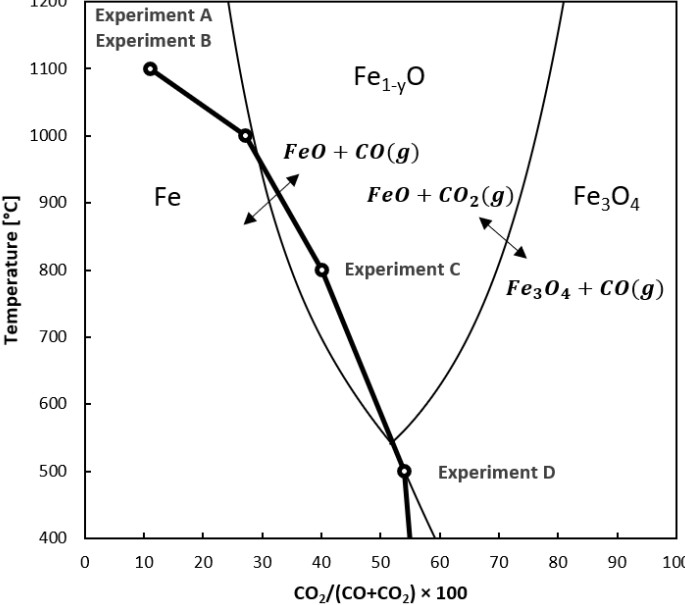

**Figure 6.** Bauer–Glaessner diagram that presents the equilibrium between iron oxides, temperature, and gas composition. The thick line represents the reducing conditions during Experiments A–D.

The dimensions of the briquette were measured, and the briquette was weighed separately with and without the sample basket. The briquette was contacted with a thermocouple that was connected to the BFS computer system. The sample basket was placed inside the reduction tube and attached to a hook that was connected to the scale of thermogravimetric analysis (TGA) to continuously measure the weight of the briquette sample during the experiments. The measurement accuracy was improved by using the scale tool in the computer program that detected a possible leaning of the sample basket against the wall of the reduction tube. Once the sample basket was in a good position, the openings of the lid were covered, and the lid was attached to the tube.

The reduction experiments were started by turning on the heating function. The briquettes were viewed with the camera during the experiments, and images were taken initially every 100 °C, and after the temperatures above 700 °C, every 50 °C degrees. At the end of each reduction experiment, the sample was cooled with $N_2$ gas for 20 min utilizing the connection on the tube lid.

### 2.7. Reduction and Swelling Calculation

The reduction degree (RD) for the carbon-free iron ore pellets was calculated according to ISO 7215, but the calculation was not used for the briquettes due to the inaccuracy caused by coal gasification. The RD is calculated as follows:

$$R_{final} = \left[ \frac{\Delta m}{m_0(0.430 w_2 - 0.111 w_1)} \times 100 \right] \times 100 \tag{1}$$

where $R_{final}$ is the final degree of reduction expressed as a percentage by mass; $m_0$ is the mass of the test portion before reduction; $\Delta m$ is the mass loss of the test portion after reduction; $w_1$ is the iron (II) oxide content (wt.-%) of the test portion prior to the test, determined in accordance with ISO 9035, and it is calculated from the iron (II) oxide content by multiplying it by the oxide conversion factor FeO/Fe (II) = 1.286; and $w_2$ is the total iron content of the test portion prior to the test [23].

The swelling behavior of the briquettes could be observed by calculating the change in volume based on the measurements of the external dimensions before and after the experiments. The swelling of the pellets was not studied. The swelling index (%) can be calculated as follows:

$$\Delta V_{briq.} = \frac{(A_1 h_1 - A_0 h_0)}{A_0 h_0} \times 100 \tag{2}$$

where $\Delta V_{briq.}$ is the swelling percentage expressed as a percentage by volume, $A_0$ is the surface area of the end of the cylindrical briquette sample before reduction, $A_1$ is the surface area of the end of the cylindrical briquette sample after reduction, $h_0$ is the height of the briquette before reduction, and $h_1$ is the height of the briquette after reduction [28].

### 2.8. Mineralogical Characterization

A Digital Olympus DSX1000 light optical microscope (LOM) and Zeiss Sigma field emission scanning electron microscope (FESEM) located at the University of Oulu were used in studying the phase transformations occurring in the briquette samples during the reduction experiments. Furthermore, an energy-dispersive X-ray spectroscopy (EDS) elemental analyzer was utilized. All five samples, four of which were tested in the BFS, and one was raw, were dry cut into smaller pieces, treated with epoxy, and polished in order to obtain suitable polished sections. The samples were coated with platinum for FESEM in order to detect the actual amount of carbon that could be hampered by a carbon coating.

## 3. Results

### 3.1. Strength Tests

The following strength test results were obtained from the AMCOM GROUP LLC laboratory. The crushing strength was tested for seven briquette samples 67–81 mm in length. The average crushing strength in normal conditions was 24.64 kg/cm as seen in

Table 4. The briquettes used in the mechanical drop strength test were sieved after the test, and 98.5% of the briquettes were more than 5 mm in size. The abrasion strengths for briquette fractions examined after 25, 50, 100, and 200 revolutions, defined as portions exceeding 5 mm in size, were 93%, 89%, 80%, and 64%, respectively. The abrasion strength test results are seen in Table 5.

**Table 4.** Mechanical crushing strength test results with calculated averages (AVG) and standard deviations (SD).

|  | Length (mm) | Force (kg) | Strength (kg/cm) | Strength (kg/cm$^2$) |
|---|---|---|---|---|
|  | 77.0 | 183.0 | 23.8 | 196.3 |
|  | 81.0 | 192.0 | 23.7 | 195.4 |
|  | 75.0 | 184.0 | 24.5 | 202.1 |
|  | 68.0 | 170.0 | 25.0 | 206.2 |
|  | 73.0 | 189.0 | 25.9 | 213.6 |
|  | 73.0 | 198.0 | 27.1 | 223.5 |
|  | 67.0 | 151.0 | 22.5 | 185.6 |
| AVG | 73.4 | 181.0 | 24.6 | 203.2 |
| SD | 4.9 | 15.9 | 1.5 | 12.6 |

**Table 5.** Results from abrasion strength test carried out in a rotating drum.

| Number of Revolutions | Content (%) | |
|---|---|---|
|  | >5 mm | >0.5 mm |
| 25 | 93 | 95 |
| 50 | 89 | 90 |
| 100 | 80 | 82 |
| 200 | 64 | 69 |

*3.2. Reduction Experiments for Auger Pressing Briquettes*

The relative weight losses that occurred during the Experiments A–F are shown in Figure 7. Swelling of the auger pressing briquettes and the reference briquette during Experiments A–D and F are shown in Figure 8.

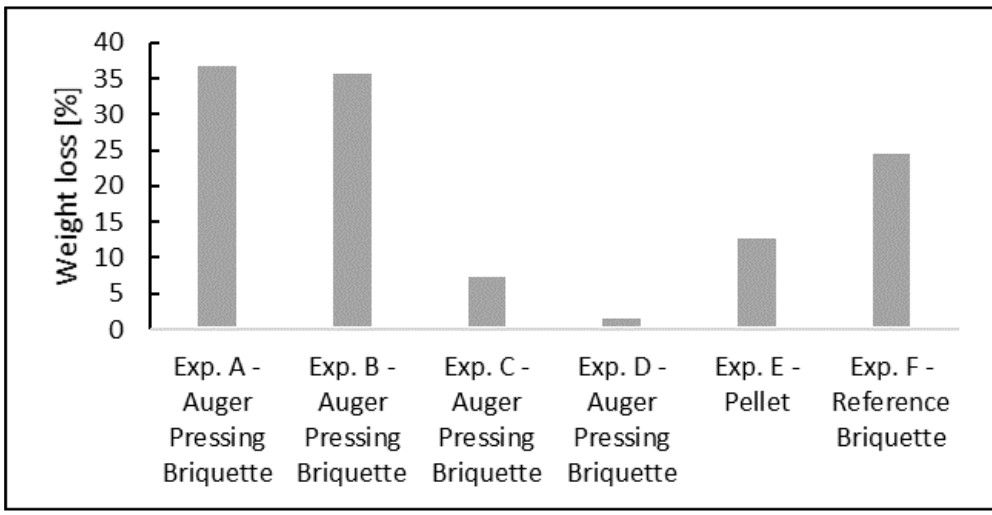

**Figure 7.** Weight losses (%) of the auger pressing briquettes (Experiments A–D), the pellets (Experiment E), and the reference briquette (Experiment F).

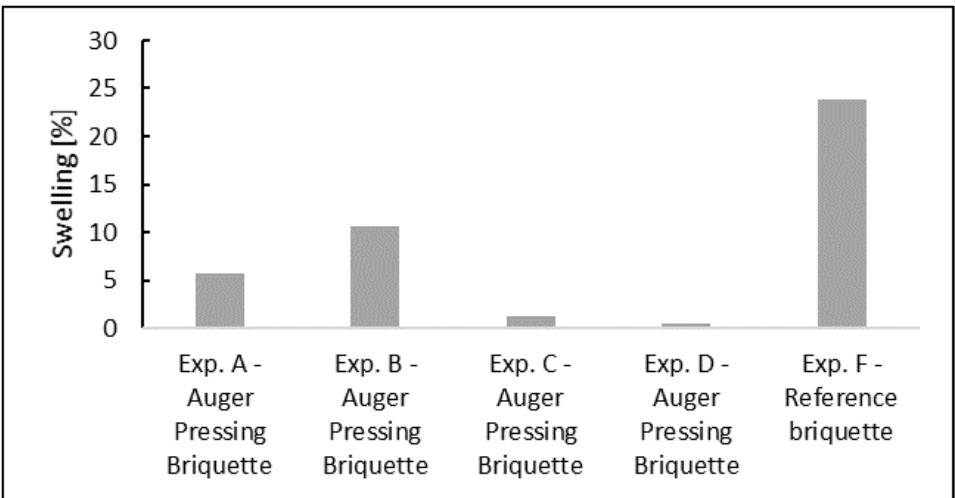

**Figure 8.** Swelling (%) of the auger pressing briquettes (Experiments A–D) and the reference briquette (Experiment F). Swelling of the pellets (Experiment E) was not studied.

Based on Figures 7 and 8, weight loss appears to be most significant in the case of auger pressing briquettes during Experiments A and B, and the reference briquette swelled more sensitively than the auger pressing briquette. TGA data provide more detailed information on the time of the weight losses occurred. The weight change curves for each experiment as a function of time together with the temperature curve are demonstrated in Figure 9.

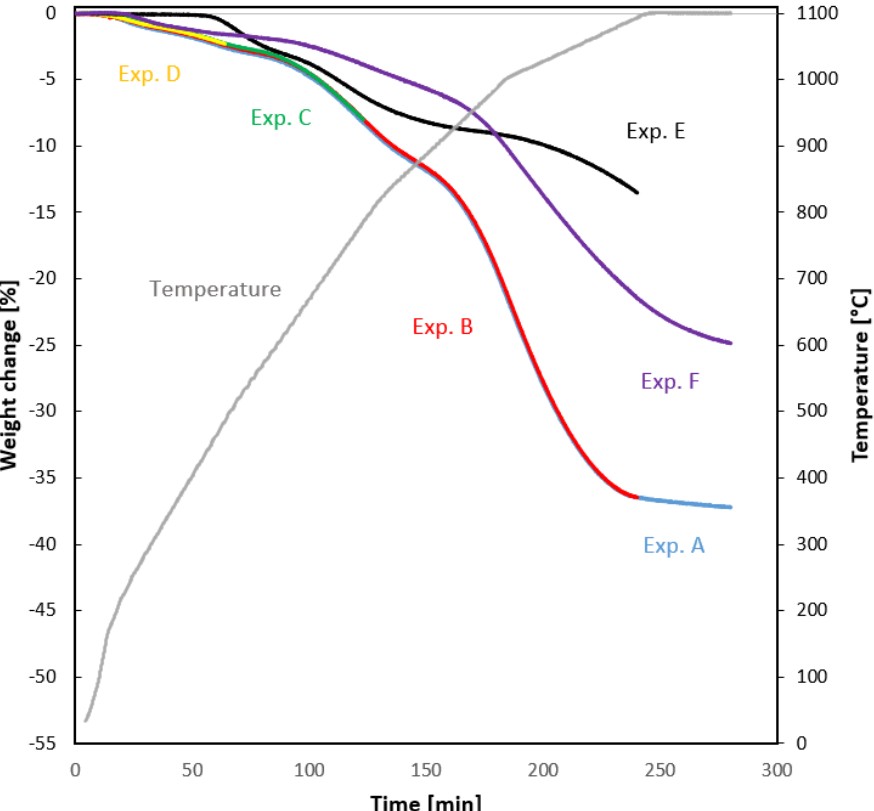

**Figure 9.** Weight change (%) curves as a function of time in the reduction experiments for the auger pressing briquettes (Experiments A–D), the pellets (Experiment E), and the reference briquette (Experiment F).

### 3.2.1. Uninterrupted Reduction Experiments

In uninterrupted experiments, referred to as Experiments A and B, reduction was performed up to the iron segment through all reduction stages presented in more detail below. As the last chemical reaction in the experiment, wüstite was reduced to metallic iron according to Equation (3).

$$FeO + CO(g) = Fe + CO_2(g) \tag{3}$$

As seen from Figure 7, the relative weight losses during Experiments A and B are almost equal: 37.0% and 35.9%. The weight loss curves for auger pressing briquettes seen in Figure 9 overlap almost completely. It can be concluded that the 40 min isothermal period did not have a major effect on weight change, although the weight decreased slightly during it. Thus, the iron contained in the briquette can be assumed to be completely reduced (RD = 100%) and the rest of the weight loss to be due to coal gasification.

At about 1000 °C, the precipitation of a white substance was observed on the glass of the lid of the reduction tube and on the stems of the sample basket. This was apparently zinc excretion. As mentioned above, the circulation behavior of zinc with a quite low boiling point of 908 °C is known to be a nuisance in the blast furnace process. Zinc is likely to be present as a compound, such as zinc oxide (ZnO). Of the samples used, the phenomenon occurred only in the case of auger pressing briquettes with a zinc content of 0.17 wt.-%. Other samples contained significantly less zinc.

The swelling of the briquettes was slight. The results for Experiment A and B were 5.8% and 10.7%, and the swelling behavior was difficult to detect visually. Its occurrence could not be detected during the reduction tests despite the camera running continuously. Instead of swelling, the briquette seemed to roll in the basket due to the rise in temperature. Minor changes were observed in the external dimensions of the briquette samples as they were measured before and after the experiments. However, the briquette samples seen in Figure 10 show similar cracking behavior during Experiments A and B, indicating that no significant cracking or other external changes occurred during the 40 min isothermal period at 1100 °C.

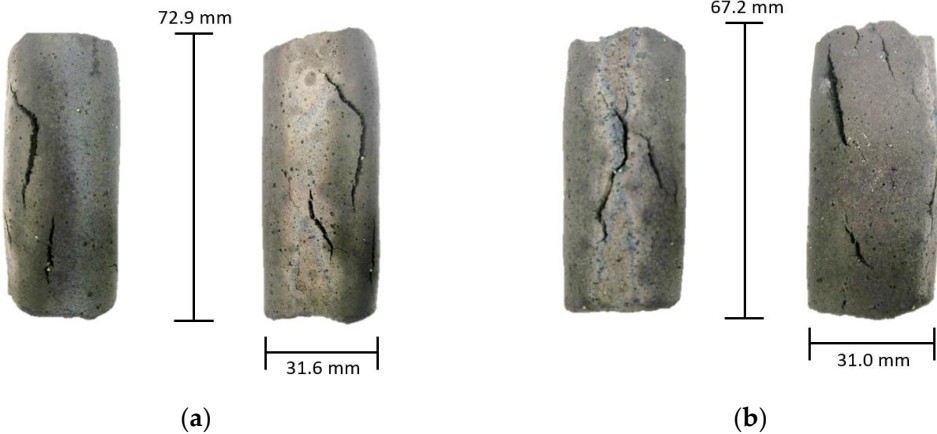

**Figure 10.** Cracks formed in the auger pressing briquette sample during (**a**) Experiment B and (**b**) Experiment A, i.e., the run with the isothermal period of 40 min.

### 3.2.2. Interrupted Reduction Experiments

In interrupted Experiments C and D, the relative weight losses were lower: 7.5% and 1.8%. Based on the weight losses, the reduction was very slow at these reduction stages, although it was faster with the auger pressing briquettes than with the reference samples. At first, reduction of hematite to magnetite occurred in both experiments according to Equation (4).

$$3Fe_2O_3 + CO(g) = 2Fe_3O_4 + CO_2(g) \tag{4}$$

In Experiment C, the reduction of magnetite to wüstite was achieved according to Equation (5).

$$Fe_3O_4 + CO(g) = 3FeO + CO_2(g) \tag{5}$$

The swelling observed during Experiments C and D, about 1.2% and 0.5%, respectively, was so minor that the differences in measurements could be due to measurement inaccuracies. The measurement was hampered by the uneven shape of the briquette ends. No visually detectable changes were observed, except for a slight discoloration of the briquette in Experiment C. That is, as a result of magnetite being reduced to wüstite.

### 3.3. Reduction Experiments for Reference Samples

In Experiments E and F, the relative weight losses for the iron ore pellets and the reference briquette were 13.0% and 24.8%, respectively. It can be seen from Figure 9 that the reduction behavior differs from auger pressing briquettes. The pellets began to lose weight at just below 500 °C, which is later than for the other samples. At this point, it can be assumed that the reduction of hematite to magnetite occurred. The pellet started to react more effectively when the temperature approached 1100 °C. The weight losses that occurred in the beginning of the Experiments A–D and F are discussed in more detail in Section 4. The RD was calculated for the iron ore pellet based on the weight loss in Experiment E by utilizing Equation (1) and the chemical analysis of the pellet shown in Table 2. The RD of the pellet was 45.3%. No cracking was observed in the pellets.

Compared to the auger pressing briquette, the reference briquette lost relatively more weight during the isothermal period although the overall weight loss was smaller. The carbon content is lower than in the auger pressing briquette, so the sample apparently was not completely reduced. Weight loss was greatest above 950 °C. Figure 11 shows the swelling behavior of the reference briquette. The most significant swelling took place above 1000 °C. More swelling occurred than in the case of the auger pressing briquette: 23.7%. No cracking was observed. The results are in line with a similar study on punch-and-die industrial briquettes carried out by Kemppainen et al. [28].

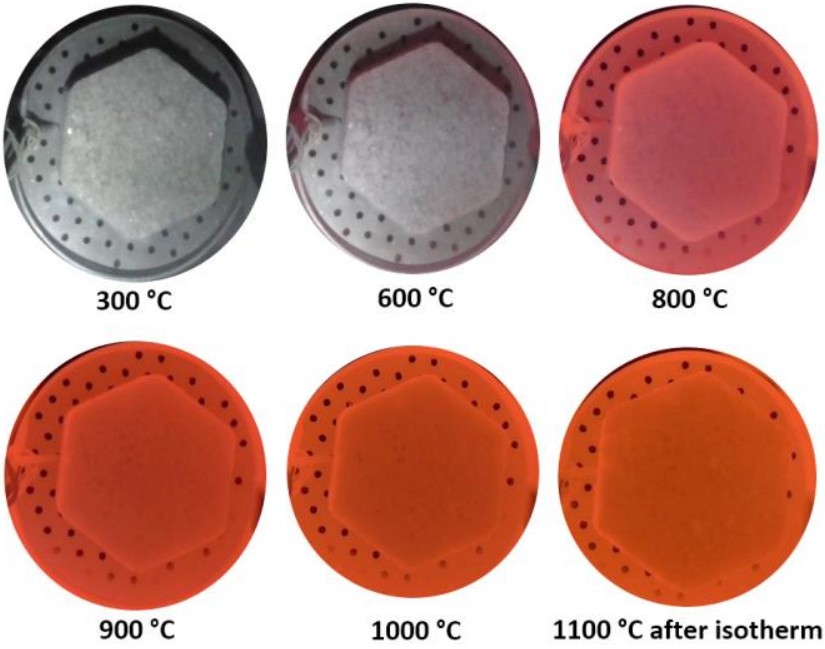

**Figure 11.** Swelling of the reference briquette with increasing temperature during Experiment F.

## 4. Discussion

### 4.1. Weight Loss and Reduction

The results show that a strong self-reduction reaction has taken place in the auger pressing briquette samples during the reduction experiments. The auger pressing briquettes contained the most carbon, i.e., 9.87 wt.-%, and the weight losses occurred considerably faster than in the reference samples. The carbon content of the reference briquette was 8.28 wt.-% and the pellet was of carbon. The reproducibility of the experiments was good, as the weight loss curves seen in Figure 9 overlapped. After a sharp weight loss phase between temperatures of 900 and 1050 °C, the curve rapidly subsided. According to the literature, coal gasification or the Boudouard reaction, which has been found to control the reduction rate [29], occurs at temperatures above 800 °C and produces CO that increases the reduction rate of the briquette [30]. The Boudouard reaction is shown in Equation (6).

$$CO_2(g) + C = 2CO(g) \tag{6}$$

In a study by Liu et al. [30], direct iron ore reduction with coal was investigated and it was found that the rate of reduction is lower between 740 and 800 °C and higher between 800 and 870 °C when magnetite is reduced to wüstite. In experiments on auger pressing briquettes, the most significant reduction appears to have started at temperatures above 900 °C. The almost complete cessation of weight loss within the period studied indicates that the sample has fully reacted. Adequate reducibility is one of the properties of a good BF charge material.

However, as mentioned above, reduction calculations for the briquette samples were not possible due to the carbon they contained in the form of coke. The coal gasification partially caused weight losses during the experiments and carbon residues were still present in the samples after the experiments. This was confirmed by microscopic images. Light optical microscope images seen in Figure 12 show how the areas with coke and iron change with reduction during Experiments A–C. In the raw sample, iron ores are found as metallic iron droplets as well as areas in different degrees of reduction. At high temperatures, most of the coal gasifies away from the sample and the metallic iron remains as distinct white spots. The image taken from the briquette sample used in Experiment D, i.e., where reduction to magnetite was sought, is not much different from the image taken from the raw sample and therefore is not presented here.

Looking again at the results presented in Figure 9, it is noted that weight losses occurred in all samples except the pellets at the beginning of the experiments when the temperatures were less than 400 °C. The weight losses that took place before the reduction may partly be the consequence of the reaction called carbonatation, which is possible due to the slaked lime or portlandite used in the briquette. In the reaction, portlandite ($Ca(OH)_2$) and carbon dioxide form calcite ($CaCO_3$) and water. According to the literature, portlandite decomposes approximately between 450 and 550 °C and calcite approximately between 700 and 900 °C. Crystallization water of the hydration products disappears gradually between 100 and 1000 °C [31] but the proportion of portlandite in the briquette is so small that the reaction in this case would not explain the weight loss. In a study by Kemppainen et al. [28], X-ray diffraction (XRD) analysis showed that portlandite had disappeared from the briquette samples at temperatures below 480 °C and calcium carbonate had increased. It was reported that the calculated total decomposition of portlandite could cause about 0.3% relative weight loss to the sample. The carbonatation reaction is shown in Equation (7).

$$Ca(OH)_2 + CO_2(g) = CaCO_3 + H_2O(g) \tag{7}$$

The presence of volatiles not studied here might explain the remaining weight losses. The possible release of zinc was one of the observed phenomena.

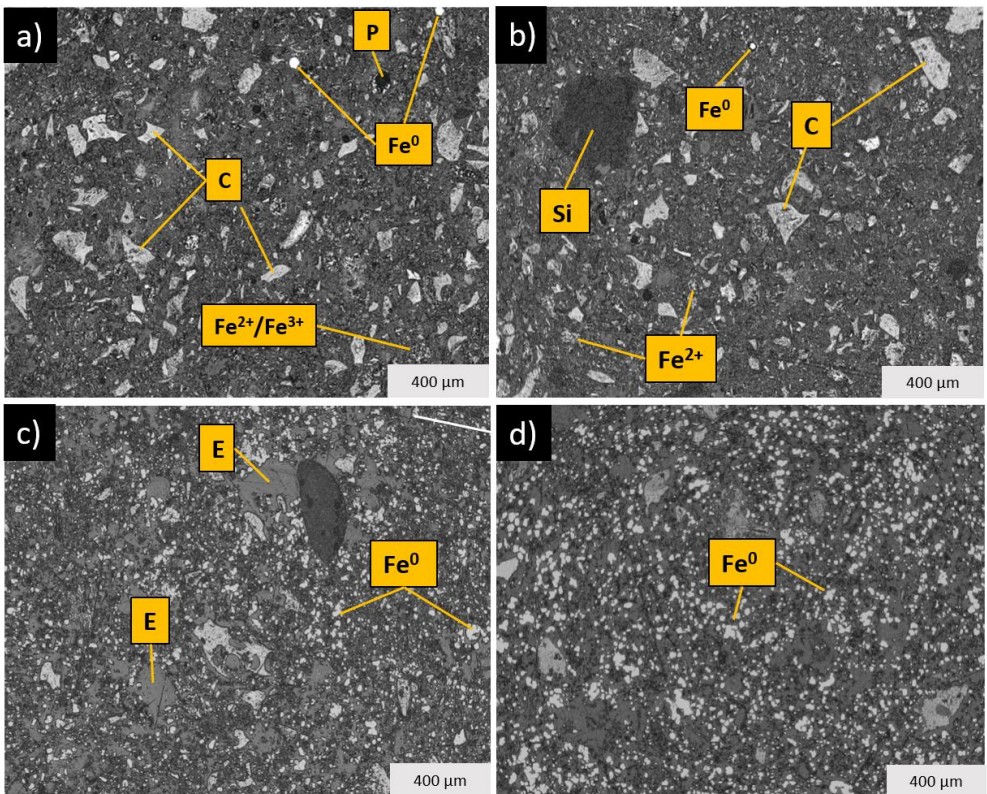

**Figure 12.** Light optical microscope (LOM) images of (**a**) raw auger pressing briquette sample, (**b**) auger pressing briquette sample after Experiment C, (**c**) auger pressing briquette sample after Experiment B, and (**d**) auger pressing sample after Experiment A ($Fe^0$ = metallic iron, $Fe^{3+}$ = trivalent iron, $Fe^{2+}$ = divalent iron, Si = silicate phase, P = pore, C = coke, E = epoxy).

## 4.2. Phase Transformations

The phase observations with LOM were evaluated using FESEM and an energy-dispersive X-ray spectroscopy (EDS) elemental analyzer. The evaluated phases of the raw sample as well as the samples treated in uninterrupted and interrupted experiments seen in Figure 13 show that the reduction reactions have occurred as expected.

The original briquette sample contained iron at all reduction stages, from which metallic iron, wüstite, and magnetite are seen in the microscopic image of the raw sample. Moreover, coal stands out as clear areas in the images. The differences between the samples from Experiment A and B are very small and possibly related only to the amount of coal. It should be noted that carbon is quite a light element, the amount of which may not be accurately identified by the EDS. According to analysis, carbon was still present in platinum-coated samples. The clear coke areas seen in the FESEM images of the raw sample and the samples from Experiments C and D have disappeared after the reduction step from wüstite to metallic iron. The image of the sample of Experiment A shows metallic iron and calcium silicate phases, i.e., iron (Fe), periclase (MgO), lime (CaO), and larnite ($Ca_2SiO_4$). The calcium carbonates still detectable in the sample of Experiment C have disappeared through the thermal decomposition at 840 °C, releasing carbon dioxide gas and calcium oxide according to Equation (8).

$$CaCO_3 = CaO + CO_2(g) \qquad (8)$$

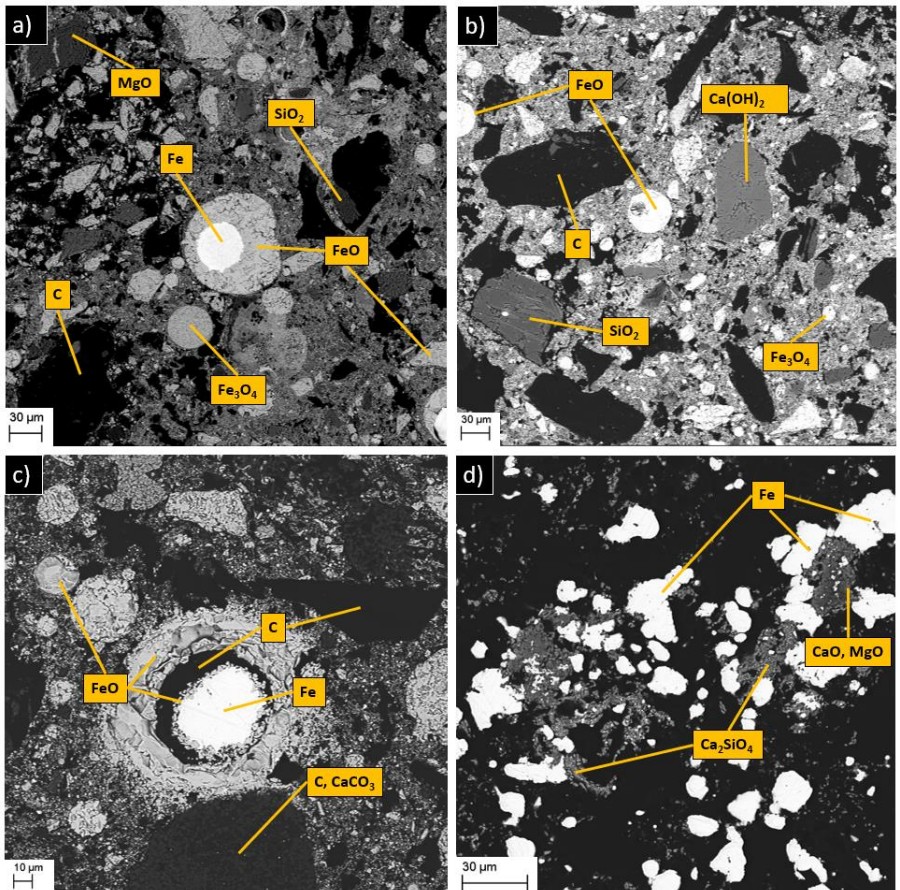

**Figure 13.** Field emission scanning electron microscope (FESEM) images and phases evaluated using an energy-dispersive X-ray spectroscopy (EDS) elemental analyzer of (**a**) the original briquette sample, (**b**) the briquette sample after Experiment D, (**c**) the briquette sample after Experiment C, and (**d**) the briquette sample after Experiment A.

### 4.3. Swelling and Cracking

The swelling of the auger pressing briquettes was very slight. The changes in the volume of the sample in the uninterrupted experiments were 5.8% and 10.7% and in the interrupted experiments practically insignificant. Due to the small amount of swelling, its time of occurrence could not be fully estimated. However, the reference briquette with a swelling result of 23.7% only began to swell significantly at temperatures around 1000 °C during the reduction step from wüstite to iron.

According to the literature, catastrophic swelling may result in a volume increase of over 300% in extreme cases. This is possible when fibrous morphology often called iron whiskers, a characteristic of pellets, is formed during that stage of the reduction process. However, it is more likely that swelling of cement-bonded briquettes occurs when reduced iron particles that comprise an individual pellet-fine particle separate from each other [32,33]. It is noteworthy that the briquette appeared to swell more without the 40 min isotherm at 1100 ° C than with it. The result may be due in part to measurement inaccuracies, but it is also in line with the study on stiff extrusion briquettes by Mohanty et al. [14] who observed a swelling index of 19% at 1000 °C but found it to drop sharply with a rise in temperature. At 1400 °C, the swelling index was only 4%. They concluded that the phenomenon was due to the sintering of iron particles which limits the growth of the fibers that cause swelling.

In terms of BF operation, such a minor swelling of the briquettes should not pose any challenges. Instead, it is worth noting that at the temperatures where swelling can be expected to occur, cracking was also observed. During Experiment B, the camera detected

a crack on the briquette at just over 1000 °C. The cracks did not seem to have expanded or increased with increasing temperature up to 1100 °C or during the isothermal period. The briquette also withstood handling without degradation after the reduction experiments. On the other hand, the reference briquette, which swelled but was not completely reduced, did not crack during the experiments.

Based on the cold strength tests and the BFS experiments, the structure of the auger pressing briquette is quite strong. It is noteworthy that the decomposition of the cement phases did not lead to the degradation of the briquette. Kemppainen et al. [28] concluded in their study on the punch-and-die briquette that phase transformation of $Ca(OH)_2$ first to $CaCO_3$ and then to $Ca_2Fe_2O_5$ may affect this behavior by reinforcing the structure with iron atoms as the reduction occurs. In order to more accurately assess whether the mechanical strength is sufficient for BF use, the reduction–softening behavior should be further studied by simulating the effect of the BF iron burden materials on the briquette during reduction. Furthermore, a dynamic low-temperature disintegration (LTD) test can be carried out for the sample. It can be used to evaluate the tendency of the agglomerate to generate fines during the reaction that occurs in the upper part of the BF shaft. The test is possible when a sample size of at least 500 g is available [34,35].

### 4.4. Effect of Chemical Composition

No chemical analysis was carried out for the auger pressing briquette due to its complex composition. The total composition for the auger pressing briquette was calculated. The contents for the briquettes studied and the reference samples are seen in Table 6 for comparison.

**Table 6.** Comparison of the chemical compositions of the samples used.

| Sample | Fe$_{tot}$ | SiO$_2$ | CaO | MgO | Al$_2$O$_3$ | Zn | H$_2$O | C | Basicity |
|---|---|---|---|---|---|---|---|---|---|
| Auger Pressing Briquette | 56.3 | 2.05 | 3.65 | 3.9 | 0.88 | 0.17 | 0.60 | 9.87 | 1.78 |
| Ref. Briquette | 48.4 | 8.1 | 11.1 | 2.1 | 2.4 | 0.01 | ~7.5 | 8.28 | 1.37 |
| Ref. Pellet | 66.7 | 1.85 | 0.43 | 1.3 | 0.32 | <0.003 | 1.5 | - | 0.23 |

As seen from Table 6, the iron content of the auger pressing briquette is not at the level of the pellet but is higher than that of the reference briquette. Moreover, the auger pressing briquette contains the most carbon. The amount of slag constituents CaO, $SiO_2$, $Al_2O_3$, and MgO or gangue content is on average higher in the reference briquette. In auger pressing briquettes, those components originate from BF sludge except for added slaked lime. The CaO/$SiO_2$ ratio or basicity is higher with the auger pressing briquettes. However, the reference briquette still contains significantly more CaO, $SiO_2$, and $Al_2O_3$. It has been found that the swelling increases with the increase of these four slag forming oxides mentioned [32]. According to contents shown in Table 6, the calculated B2 basicity (CaO/$SiO_2$ ratio) of the auger pressing briquette is 1.78, which is quite high compared to typical burden materials.

The auger pressing briquette contained zinc, while the zinc contents of the reference samples were insignificant. The zinc content of a BF sludge may be high, but this time it was only 0.43%, whereby the zinc content of the briquette became 0.17%. It may not be necessary to remove such a small amount of zinc from the briquette before BF use. Nowadays, the BF permissible zinc load is 150 g/tHM. Higher concentrations may be detrimental to BF use. Lundkvist et al. [36,37] used a novel OXYFINES technique to upgrade zinc-containing BF sludge to enable its recycling. They stated that higher temperature and lower oxidation potential enhance the zinc evaporation. BF sludge coal content can be utilized in increasing the degree of oxidation, which further raises the oxygen potential in the reaction zone. Elemental zinc vapor formation shown in Equation (9) is negatively affected by the

increased oxygen potential. Thus, it is beneficial to optimize the oxidation, which can be expressed as a $CO_2/CO$ ratio, for example.

$$ZnO + C = Zn\ (g) + CO\ (g) \tag{9}$$

The composition of the binder used in the auger pressing briquette seemed a robust combination of inorganic and organic materials. Organic binders tend to decompose at high temperatures (>400 °C) leading to briquette degradation [17] but have no side effects on iron ore grade like inorganic binders [5]. The reference briquette, which used ground-granulated BF slag as a binder, i.e., inorganic binder, did not crack. Secondly, it was not reduced as efficiently as the auger pressing briquette. In future research, different recipes should be studied in order to evaluate the effect of each by-product on the mechanical properties of the briquette, which will also facilitate the choice of binder.

## 5. Conclusions

Laboratory tests simulating actual BF conditions were carried out to evaluate the suitability of by-product-based auger pressing briquettes for BF use. Both uninterrupted and interrupted experiments in a $N_2$–CO–$CO_2$ atmosphere were performed to study phase transformations occurring under reducing conditions. In addition to reduction, the swelling and cracking behavior of briquettes was of interest. Based on the BFS experiments, the following conclusions can be drawn:

1.  BF sludge-based by-product briquettes made by the vacuum auger pressing technology had a strong self-reducing effect based on the rapid weight losses at temperatures 780–1100 °C compared to the reference samples. The phenomenon is most probably caused by the coal contained in the briquette. This is a particularly good feature in terms of BF productivity.
2.  Slight swelling behavior was observed, as the volume of the briquettes increased by 5%–11% when wüstite was reduced to metallic iron during the uninterrupted reduction experiments. Such a small increase in volume does not cause operational issues in BF.
3.  The cracking behavior of briquettes at temperatures above 1000 °C was observed, but the briquettes could be handled after the reduction experiments without degradation. Such durability is promising since it is desired to avoid fines ending up in the BF process.
4.  The chemical composition of the briquettes appears to be a good compromise between reducibility and strength properties, allowing further research into new by-product briquette recipes as well.

**Author Contributions:** Conceptualization, O.V. and M.I.; methodology, O.V. and M.I.; software, M.I.; formal analysis, O.V.; investigation, O.V., M.I., N.K. and D.S.; resources, T.F.; writing—original draft preparation, O.V.; writing—review and editing, O.V., M.I., A.H., I.T., and T.F.; visualization, O.V. and I.T.; supervision, M.I., A.H., and T.F.; project administration, T.F. and I.T.; funding acquisition, T.F. and A.M. All authors have read and agreed to the published version of the manuscript.

**Funding:** This research was funded by Business Finland as a part of the Towards Carbon Neutral Metals (TOCANEM) research program, grant number 40693/31/2020.

**Data Availability Statement:** Not applicable.

**Acknowledgments:** SSAB is acknowledged for providing the reference briquettes and the Center for Material Analysis (CMA) for providing mineralogical characterization services. Tommi Kokkonen is acknowledged for his technical support in laboratory work at University of Oulu.

**Conflicts of Interest:** Olli Vitikka, the corresponding author, acts as a doctoral researcher and Mikko Iljana and Anne Heikkilä, the co-authors, act as post-doctoral researchers at the University of Oulu and declare no personal conflict of interest. Illia Tkalenko, a co-author, acts as a technical director, Nikita Koriuchev, a co-author, acts as a laboratory director, and Daniel Shehovsov, a co-author, acts as

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
