# Peer review of "Suitability of Auger Pressing Briquettes for Blast Furnace Use Based on Laboratory Tests"

_minerals, doi:10.3390/min12070868_

Round 1

Reviewer 1 Report

In this paper, the reducibility, swelling and cracking behavior of the auger pressing briquette were analyzed by continuous high-temperature reduction experiments and interrupted high-temperature reduction experiments. In addition, the strength of the briquette was also analyzed. The feasibility of applying auger pressing briquette in blast furnace was verified. Some useful results will have benefits to the practice. It can be considered to published after the following revisions:

1. There are too many keywords, please delete some useless keywords;

2. Because the strength, swelling and cracking of the briquette are not only related to the production process of the briquette, but also related to the raw materials and binder of the briquette. Whether the raw materials and binders of the reference vibro-pressing briquette are the same as the auger pressing briquette. It is recommended to explain its raw material composition and binder in the method.

3. In Section 2.4(Strength test), There are too much irrelevant information is introduced here, but the specific method for testing the three strengths of briquette is not provided. This results in the reader not being able to comprehend your subsequent measurements of briquette strength very well. Please add the full detailed strength measurement method here.

4. In section 3.1, there are 4 blank lines in lines 287-291, please modify the format.

5. In table 4, what is the meaning of AVG and SD? Please explain their meaning in the captions.

6. In section 4.2(Phase transformation), SEM-EDS cannot accurately observe the phase transition. If possible, it is recommended to use X-ray diffractometer to analyze the phase composition to verify the phase composition of each stage.

7. In section 4.3, Where does this phraseAccording to literature, catastrophic swelling may result in a volume increase of over 300% come from? Is the value of 300% accurate? It is generally understood that 30% is a catastrophic expansion.

8. In section 4.4, How to understand The degree of oxidation is increased, and increased oxygen potential is attained in the reaction zone by the utilization of the BF sludge coal content? Why use the coal content of blast furnace sludge can increase the degree of oxidation and increase the oxygen potential in the reaction zone? Please give an explanation.

Author Response

Dear Reviewer,

Thank you for your review report. Please see the attachment for response.

- Authors

Reviewer 2 Report

The paper presents very interesting and important problem of possibility using of new kind of briquettes in blast furnace process. Main goal of the paper is the analysis of it properties, but very important element is using ferrous waste in production process of these briquettes. Authors present their main results of the analysis. It is done in interesting and understandable way. I have a few comments, but they are rather connected with improving only the manuscript (problems do not influence on the scientific level of the text):

·         References 19 & 24: lack of full descriptions of references,

·         Page 9, lines 286-291: something is wrong with the text.

I have also one question to the authors: did you analyze the economic effects of use of such materials? On one hand we have cost of production of such materials, on the other hands – savings. But of course, we have also other parameters (like efficiency of blast furnace process when such materials are used). And maybe the second: in what percentage can this briquette replace iron ore in the blast furnace process? Is it possible in 100%?

Author Response

(The authors gave the same response as above.)
